# Towards Synergistic Path-based Explanations for Knowledge Graph Completion: Exploration and Evaluation

**Tengfei Ma**[1]   **Xiang Song**[2]   **Wen Tao**[1]   **Mufei Li**[3]   **Jiani Zhang**[2]   **Xiaoqin Pan**[1]   **Yijun Wang**[1]
**Bosheng Song**[1]   **Xiangxiang Zeng**[1,*]
[1]College of Computer Science and Electronic Engineering, Hunan University
[2]Amazon AWS AI   [3]Georgia Institute of Technology
`tfma@hnu.edu.cn   xiangsx@amazon.com   xzeng@hnu.edu.cn`

## Abstract

Knowledge graph completion (KGC) aims to alleviate the inherent incompleteness of knowledge graphs (KGs), a crucial task for numerous applications such as recommendation systems and drug repurposing. The success of knowledge graph embedding (KGE) models provokes the question about the explainability: "*Which the patterns of the input KG are most determinant to the prediction*?" Particularly, path-based explainers prevail in existing methods because of their strong capability for human understanding. In this paper, based on the observation that a fact is usually determined by the synergy of multiple reasoning chains, we propose a novel explainable framework, called KGExplainer, to explore synergistic pathways. KGExplainer is a model-agnostic approach that employs a perturbation-based greedy search algorithm to identify the most crucial synergistic paths as explanations within the local structure of target predictions. To evaluate the quality of these explanations, KGExplainer distills an evaluator from the target KGE model, allowing for the examination of their fidelity. We experimentally demonstrate that the distilled evaluator has comparable predictive performance to the target KGE. Experimental results on benchmark datasets demonstrate the effectiveness of KGExplainer, achieving a human evaluation accuracy of 83.3% and showing promising improvements in explainability. Code is available at `https://github.com/xiaomingaaa/KGExplainer`.

## 1 Introduction

Knowledge graph completion (KGC) aims to infer missing facts and tackles the incompleteness of knowledge graphs (KGs) (Akrami et al., 2020), which has been widely used to support various applications including recommendation (Wang et al., 2019a; Yang et al., 2022; Wang et al., 2022), sponsored search (Lin et al., 2022), and drug discovery (Ma et al., 2023; Bang et al., 2023; Pan et al., 2022). Most KGC models utilize knowledge graph embedding (KGE) methods to map KG elements into multi-dimensional vectors (Nguyen et al., 2023; Song et al., 2023). These vectors are commonly used as features for downstream tasks and input into scoring functions for predicting missing links. KGE methods have been shown to achieve more powerful predictive performance than traditional models and can be easily scaled to large graphs (Ali et al., 2021; Zheng et al., 2020). However, current KGE methods make black-box predictions without providing explanations, hindering their deployment in risk-sensitive scenarios (Wang et al., 2022; Zhang et al., 2023).

To address the aforementioned limitation, the idea of adversarial modifications is adopted to find the key facts, which identify the facts to include or exclude from the KG and monitor their prediction score change in the perturbed KG (Pezeshkpour et al., 2018; Rossi et al., 2022; Betz et al., 2022). These methods can provide discrete evidence, which is insufficient to completely explain why the model makes such a prediction (Zhang et al., 2023). To capture the coherent reasoning chains for target predictions, some researchers extract possible logical rules by leveraging the known facts

---

*Corresponding Author

within the KG (Zhang et al., 2019; Sadeghian et al., 2019; Arakelyan et al., 2021). The mined rules are usually represented as paths, which can give the basis of decision-making and enhance the transparency of the prediction model. Besides, a line of work adopts reinforcement learning to identify the next node step-by-step and form a reasoning chain by designing a reward model (Xian et al., 2019; Bhowmik & de Melo, 2020; Jiang et al., 2023). Apart from them, more recent approaches prefer relying on graph neural networks and introducing pruning mechanisms, learning an edge scorer to search for valid paths as explanations (Zhang et al., 2023; Chang et al., 2024).

Despite the fruitful progress and the popular trend toward path-based explainers, generating multiple synergistic paths for complex queries has not been discussed. For example, as shown in Figure 1a, *Alibaba* can be identified as a competitor of *JD.com* over a complex KG[1] when given the synergistic facts: both *Alibaba* and *JD.com* invest in retail in China. In addition, for the prediction that the drug *Acetaminophen* decreases the serum concentration of drug *Warfarin* (Figure 1b), the explainers need to provide multiple synergistic paths simultaneously to describe the facts: *Acetaminophen* inhibits the activity of the enzymes that metabolize *Warfarin*, thus bringing a decrease in the serum concentration of *Warfarin*.

Additionally, for the identified explanations, we cannot directly verify their rationality due to the ground truth is unavailable. Previous methods focused more on model faithfulness (Pezeshkpour et al., 2018; Xian et al., 2019) or manual verification of small-scale facts case by case (Zhang et al., 2021), ignoring the quantitative assessment of model explainability.

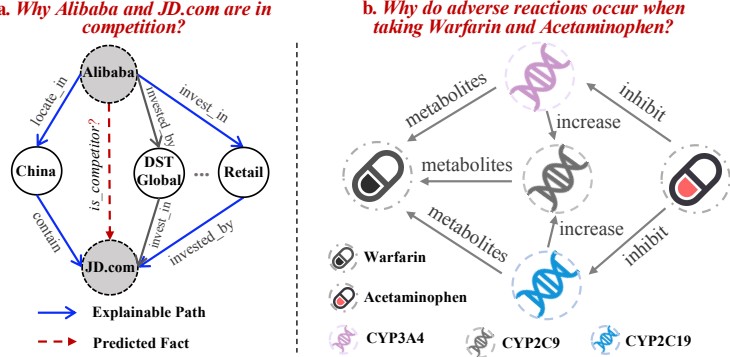

Figure 1: (a) Multiple paths (blue) explain the competition relationship between *Alibaba* and *JD.com*; (b) Identifying the adverse reactions between *Warfarin* and *Acetaminophen* needs the synergy of multiple paths to explain the interaction mechanism.

Based on the above observations, we propose KGExplainer to explore synergistic path-based explanations for KGC tasks and provide an effective evaluation. KGExplainer is a post-hoc explainable model performed on the trained KGEs, which searches for single or multiple path-based reasoning chains. We first perform a model-agnostic greedy search approach to identify important paths and distill an explanation evaluator from target KGEs to measure the correlation between explanations and predictions. Then we evaluate the fidelity (Wu et al., 2023) of the identified synergistic paths by measuring whether the original prediction can be maintained based on the explored explanations.

To the best of our knowledge, we are the first to approach the explanations of KGE-based KGC models from a novel perspective by emphasizing and evaluating synergistic paths. Besides, we propose an evaluation strategy to quantitatively assess the effectiveness of synergistic path-based explanations by distilling an evaluator to examine their fidelity. Extensive experiments and human evaluation on benchmarks demonstrate the effectiveness and efficiency of KGExplainer.

## 2 RELATED WORKS

**Knowledge Graph Completion.** KGC aims to address the invariable incompleteness of knowledge graphs (KGs) by identifying missing interactions between entities. Previous rule-based methods (Sadeghian et al., 2019; Arakelyan et al., 2021) mine logical rules iteratively based on pre-trained embeddings of KGEs during the training process. These methods provide reasoning chains for prediction, which enhance the model's transparency. Knowledge graph embedding (KGE) models map KG elements to multi-dimensional vectors and define a scoring function to infer new

---

[1]Alibaba and JD.com are online retail companies in China.

facts, showing superior performance and scalability over rule-based methods for predicting missing links (Zhang et al., 2022; Wang et al., 2023; Gregucci et al., 2023). Specifically, the translational models, TransE (Bordes et al., 2013a) and its extensions (Wang et al., 2014; Lin et al., 2015), represent relations and entities as embedding vectors and treat the relation as the translation from head entities to tail entities. In another aspect, the bilinear models, RESCAL (Nickel et al., 2011) and its variants (Trouillon et al., 2016; Yang et al., 2015), represent relations with matrices and combine the head and tail entities by sequentially multiplying head embedding, relation matrix, and tail embedding. To effectively infer various relation patterns or properties (e.g., symmetry and inversion) over complex KGs, RotatE (Sun et al., 2019) represents each relation as a rotation operation from the source entity to the target entity in the complex vector space. Although these KGE models are successful in predicting unknown facts, they are limited in lacking transparency, blocking researchers from developing trustworthy models. To address the above limitations, we propose KGExplainer to enhance the explainability of KGE-based KGC models.

**Explainability in KGC.** To increase the transparency of models in KGC tasks, researchers have developed various explainable models (Wang et al., 2022; Huang et al., 2022; Zhang et al., 2023; Yao et al., 2023). Learning rules as the explanations from KGs has been studied extensively in inductive logic programming (Galárraga et al., 2013; Qu et al., 2021; Ho et al., 2018). To improve the explainability of KGE methods, CRIAGE (Pezeshkpour et al., 2018), Kelpie (Rossi et al., 2022), and KE-X (Zhao et al., 2023) search for the isolated key edges to explain the target prediction by approximately evaluating the impact of removing an existing fact from the KG on prediction score. However, discrete edges can not form a complete reasoning chain. To explore path-based explanations, PGPR (Xian et al., 2019) and ELEP (Bhowmik & de Melo, 2020) have adopted a reinforcement learning approach to predict tail entities and take the reasoning path between the head and tail entities as the evidence. PaGE-Link (Zhang et al., 2023) proposed a path-based explainable method for the graph neural network-based KGC tasks, providing valid reasoning chains. To provide enough evidence for the decision-making complex scenario, we propose KGExplainer to explore synergistic paths and form a connected subgraph as an explanation.

## 3 PRELIMINARY

**KGE-based KGC.** We define a knowledge graph (KG) as a labeled directed graph $\mathcal{G} = (\mathcal{V}, \mathcal{R}, \mathcal{E}) = \{\langle h, r, t \rangle \,|\, h, t \in \mathcal{V}, r \in \mathcal{R}, (h, t) \in \mathcal{E}\}$ where each triple represents a relation $r$ between the head entity $h$ and tail entity $t$. Most KGs are incomplete. Knowledge graph completion (KGC) leverages existing facts in the KG to infer missing ones. Currently, the KGC method adopts deep learning techniques to learn the embeddings of KG elements. Intuitively, knowledge graph embeddings (KGEs) contain the semantic information of the original KG and can be used to predict new links. Generally, the KGE-based models define a score function $\phi$ to estimate the plausibility of triples and optimize the vectorized embeddings by maximizing the scores of existing facts. For the incomplete triple $\langle h, r, ? \rangle$, the KGE models find the missing tail entity $t$ with best scores as follows:

$$t = \underset{v \in \mathcal{V}}{\operatorname{argmax}} \, \phi(h, r, v). \tag{1}$$

The head entity prediction for the unknown links like $\langle ?, r, t \rangle$ can be defined analogously. In the following sections, we mostly refer to tail entity prediction for simplicity. All our methods can be applied to head entity prediction.

**Enclosing Subgraph.** Based on GraIL (Teru et al., 2020), when given a KG $\mathcal{G}$ and a link $\langle h, r, t \rangle$ for tail prediction, we extract an enclosing subgraph surrounding the target link to represent the relevant patterns for model's decision. Initially, we obtain the $k$-hop neighboring entities $\mathcal{N}_k(h) = \{s|d(h, s) \leq k\}$ and $\mathcal{N}_k(t) = \{s|d(t, s) \leq k\}$ for both $h$ and $t$, where $d(\cdot, \cdot)$ is the shortest path distance between the entity pair on $\mathcal{G}$. We then obtain the set of entities $V = \{s|s \in \mathcal{N}_k(h) \cap \mathcal{N}_k(t)\}$ as vertices of the enclosing subgraph. Finally, we extract the edges $E$ linked by the set of entities $V$ from $\mathcal{G}$ as the $k$-hop enclosing subgraph $g = (V, E)$.

## 4 THE KGEXPLAINER FRAMEWORK

**Problem Formulation.** Given a fact $\langle h, r, t \rangle$, KGExplainer investigates the explanation that intuitively consists of the most critical synergistic path patterns featuring $h$ that allow KGE mod-

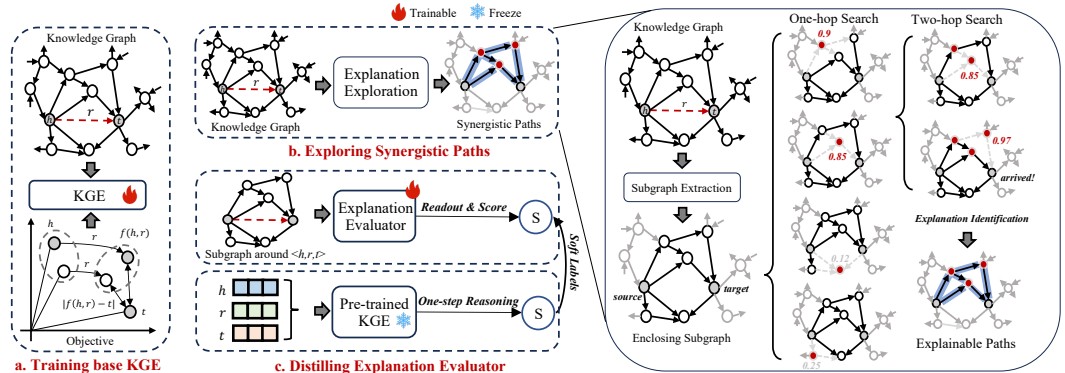

Figure 2: The KGExplainer framework comprises three modules: (a) Pre-training KGE models on the target KG ; (b) Exploring synergistic path-based explanations for pre-trained KGE models by greedy search ; (c) Distilling an evaluator from pre-trained KGE to assess the fidelity of explored explanations quantitatively.

els to predict tail $t$. For instance, when explaining why the top ranking tail for the missing fact $\langle Donald\_Trump, nationality, ? \rangle$ is $USA$, KGExplainer searches for the smallest key paths between *Donald_Trump* and *USA*, allowing the model to predict the same ranking of *USA*. Specifically, KGExplainer aims to find a connected subgraph (i.e., multiple synergistic paths) $g_{key}$ that is most influential to the target prediction $\langle h, r, t \rangle$, and removing the other irrelevant facts do not affect the model's prediction. We can define this process as follows:

$$g_{key} = \underset{\tilde{g} \in g_{sub}}{\arg\max} D(g, \tilde{g}) \tag{2}$$

$$\text{subject to } \Delta(f(g), f(\tilde{g})) = \min_{\hat{g} \in g_{sub}} \Delta(f(g), f'(\hat{g})) \tag{3}$$

where $g$ is the original enclosing subgraph , $g_{sub}$ denotes the all possible connected subgraphs from $h$ to $t$ within $g$. $f(\cdot)$ and $f'(\cdot)$ represent the original and retrained KGE models. $D(g, \tilde{g})$ represents the structure similarity between $g$ and $\tilde{g}$, and also indicates the sparsity of explanations (see Section 5.4).

### 4.1 Pre-training KGE.

In this paper, we developed the KGExplainer as a post-hoc explanation method for KGE-based KGC models by exploring synergistic paths (i.e., connected subgraph). To effectively learn semantic knowledge within the complex KG, we adopt the state-of-the-art RotatE (Sun et al., 2019) as the target KGE. Specifically, given a triple $\langle h, r, t \rangle$, we adopt a score function and expect that the embedding of head entity $h$ is similar to the embedding of tail entity $t$ by rotating it over the relation $r$. By maximizing the scores of positive triples and minimizing the scores of negative ones, we can obtain the pre-trained entity and relation embeddings (Figure 2a). Theoretically, KGExplainer is model-agnostic and can be applied to any KGE-based model with a score function. We designed experiments to verify the effectiveness of KGExplainer on various KGE models, the details can be found in Section 5.3.

### 4.2 Explanation Exploration.

**Greedy Search for Key Synergistic Paths.** As shown in Figure 2b, given a predicted link $\langle h, r, t \rangle$ for the prediction of tail entity $t$, we developed a greedy search algorithm based on a perturbation mechanism to search for crucial paths as the explanations. We first extract the enclosing subgraph $g$ around the target link, which contains informative interactions for such prediction. To identify the most important paths influencing the prediction within $g$, we search all entities in $g$ and measure their importance to capture key ones. Specifically, we remove each entity and its adjacent from $g$ and retrain[2] the target KGE on the perturbed subgraph $g'$. Then we measure the score change of

---

[2]Retraining KGE over the entire KG is way too costly and we discuss an efficient strategy to tackle this issue in Section 4.2

the target link using the original and retrained embedding as the entity importance. The more the score changes, the more critical an entity is. To ensure that the key entities obtained from the $k$-hop enclosing subgraph $g$ lead to a single connected component, we search the key entities from the head $h$ and tail $t$ hop by hop. We first remove each 1-hop neighbor entity of $h$ and measure their importance to identify the top important entities. After getting the top-$n$ important entities $\mathrm{v}_1^n$ from 1-hop neighbor entities, we then search the one-hop neighboring entities of $\mathrm{v}_1^n$ analogously for important two-hop neighbors $\mathrm{v}_2^n$. We repeat this one-hop search process until reaching the tail entity $t$ of the target link. As a result, we capture a set of the $k$-hop key neighboring entities $\mathrm{v}^n = \left\{\mathrm{v}_1^n, \cdots, \mathrm{v}_k^n\right\}$ and construct a connected subgraph for multiple synergistic paths as the explanation. We provide a detailed Algorithm in Appendix A.1.

**Retraining Strategy for Importance Assessment.** To effectively obtain the score changes of target facts based on the KGE models, we designed a retraining strategy to capture the new embedding on the perturbed KG. Since each entity perturbation only makes minor changes over the entire KG, retraining the embeddings over the revised KG is infeasible when $\mathcal{G}$ has millions of entities and facts. To focus on the relevant patterns and reduce the overhead of the retraining, we introduce a subgraph-based retraining strategy. We extract the enclosing subgraph $g$ of the target fact $\langle h, r, t \rangle$ and expand $g$ by including the 1-hop neighbors $\mathcal{N}_{extend}$ of the existing entities, which constructs a new subgraph $g_{extend}$. Then we retrain the embeddings of the KG elements on $g_{extend}$ to make the calculation of score changes effective. To keep the overall entity and relation embeddings of $g$ within the same representation space of $\mathcal{G}$, we alleviate the following constraints: (1) We initialize the embeddings of the entities in $g_{extend}$ with the pre-trained embeddings from $\mathcal{G}$; (2) We fix the relation embeddings in $g_{extend}$ as the same as those in $\mathcal{G}$; (3) We fix the embeddings of entities $v \in \mathcal{N}_{extend}$. By using the retraining strategy, KGExplainer can effectively capture the target fact's score change and efficiently search for key subgraphs. More details can be found in Appendix A.2.

### 4.3 EVALUATOR DISTILLATION

**Why Subgraph Evaluator?** Evaluating the quality of identified synergistic paths is essential. However, the ground truth explanations are unavailable, making it difficult to achieve that objective. To evaluate the quality of explanations, we introduce a hypothesis that a perfect explanation should retain the key context of target prediction and can still maintain the predicted score (Wu et al., 2023). Thus, an effective idea is to verify that the explored explanations can still obtain the target prediction. However, the base KGEs implicitly have access to the entire graph and only score one triple, which can not compute the prediction based on multiple synergistic paths. To address this limitation, we design a subgraph evaluator by distilling a relational graph neural network (GNN) from pre-trained KGEs for KGC. Given a pre-trained KGE-based KGC model $\phi(h, r, t)$, we aim to train a subgraph evaluator to replicate the predictive performance of the target KGE and evaluate the subgraph structures of explanations. During training, the GNN is optimized to predict the facts from their complete enclosing subgraphs. Once trained, the explored explanations as connected subgraphs can be fed into the evaluator and evaluate the fidelity of multiple synergistic paths (i.e., the connected subgraph).

**How to get Subgraph Evaluator?** Given a link $\langle h, r, t \rangle$, we develop a score function $Z(h, r, t)$ with a $L$-layer relational graph attention neural network (R-GAT) on its enclosing subgraph $g$ to score it based on the local substructure representation. The update function of the entities in the $l$-th layer is defined as:

$$\mathbf{x}_i^l = \sum_{r \in \mathcal{R}} \sum_{j \in \mathcal{N}_r(i)} \alpha_{(i,r,j)} \mathbf{W}_r^l \Phi(\mathbf{e}_r^{l-1}, \mathbf{x}_j^{l-1}),$$

$$\alpha_{(i,r,j)} = \text{sigmoid}\left(\mathbf{W}_1 \left[\mathbf{x}_i^{l-1} \oplus \mathbf{x}_j^{l-1} \oplus \mathbf{e}_r^{l-1}\right]\right),$$

(4)

where $\mathbf{e}_r^l$ and $\mathbf{x}_i^l$ represent the $l$-th layer embeddings of relation $r$ and entity $i$. $\alpha_{(i,r,j)}$ and $\mathcal{N}_r(i)$ denote the weight and neighbors of entity $i$ under the relation $r$, respectively. $\oplus$ is the concatenation operation. $\mathbf{W}_r^l$ represents the transformation matrix of relation $r$, and $\Phi$ is the aggregation operation to fuse the hidden features of entities and relations. Finally, we obtain the global representation $\mathbf{X}_{sub}$ of the subgraph $g$ as follows:

$$\mathbf{X}_{sub} = \frac{1}{|V|} \sum_{i \in V} f(\mathbf{x}_i^L),$$

(5)

where $V$ is the entity set of the enclosing subgraph $g$. After obtaining the global representation $\mathbf{X}_{sub}$, we define a linear layer to score the target link as follows:

$$Z(h, r, t) = \mathbf{W}^T \mathbf{X}_{sub}, \tag{6}$$

where $\mathbf{W}^T$ is a transform matrix.

To optimize the subgraph evaluator, inspired by Gou et al. (2021), we adopt the insights of knowledge distillation and consider the predictions of the pre-trained KGE-based KGC model as soft labels. Meanwhile, to enhance the model's ability to rank candidate entities, we introduce a regularization term that encourages maximizing the margin between the scores for positive and negative samples. The objective for the optimization of this evaluator is defined as follows:

$$\min \sum_{(h,r,t) \in \mathcal{G}} \|\phi(h, r, t) - Z(h, r, t)\|^2 + \lambda L(h, r, t), \tag{7}$$

where the first term is a score alignment objective. The second term is a pair-wise loss , which is defined by negative sampling:

$$L(h, r, t) = -Z(h, r, t) + \frac{1}{N} \sum_{n=1}^{N} Z(h, r, v_n), \tag{8}$$

where $N$ denotes the number of negative samples and $v_n$ is uniformly drawn from $\mathcal{V}$. By minimizing the above optimization objective, we can obtain an effective subgraph scoring function with powerful predictive performance and evaluation capabilities for synergistic path-based explanations.

### 4.4 SCALABILITY OF KGEXPLAINER

The computation complexity of finding explanations using KGExplainer is related to two parts: 1) the size of enclosing subgraph $g$ and 2) the number of entities that require importance evaluation.

**Proposition 1** *Given $\mathcal{G} = (\mathcal{V}, \mathcal{R}, \mathcal{E})$, a predicted fact $\langle h, r, t \rangle$, the maximum length of paths $L$, and the number of paths $N$, the size of $g$ is $O(\frac{LN|\mathcal{V}|}{|\mathcal{E}|})$ and the size of $\mathcal{N}_{extend}$ is $O(LN)$.*

Given $h$ and $t$, the maximum number of edges within paths is $L * N$. The probability of an entity $v \in \mathcal{V}$ in an edge $e \in \mathcal{E}$ is $\frac{2|\mathcal{V}|}{|\mathcal{E}|}$. Thus, the size of $g$ is bounded by $LN * \frac{2|\mathcal{V}|}{|\mathcal{E}|}$, i.e., $O(\frac{LN|\mathcal{V}|}{|\mathcal{E}|})$. The number of entities in $\mathcal{N}_{extend}$ is $\frac{2LN|\mathcal{V}|}{|\mathcal{E}|} * \frac{2|\mathcal{E}|}{|\mathcal{V}|}$, i.e., $O(LN)$.

**Proposition 2** *Given $\mathcal{G} = (\mathcal{V}, \mathcal{R}, \mathcal{E})$, a predicted fact $\langle h, r, t \rangle$, the maximum length of paths $L$ and the maximum number of entities per hop in an explanation $K$. The maximum number of entities to evaluate is $(1 + (L - 2)K)(\frac{2|\mathcal{E}|}{|\mathcal{V}|})$.*

Following Algorithm 1 in Appendix A.1, the average number of entities to evaluate is $d$ in the first hop, 0 in the last hop, and $K * d$ in the intermediate hops, where $d = \frac{2|\mathcal{E}|}{|\mathcal{V}|}$ is the average degree of an entity. Thus, the total number of entities to evaluate is $(1 + (L - 2)K)(\frac{2|\mathcal{E}|}{|\mathcal{V}|})$.

Here, we conclude that the average computation complexity of finding the explanation for a predicted fact $\langle h, r, t \rangle$ is only related to the maximum length of paths $L$, the maximum number of paths $N$, and the average degree of an entity in KG, which is not related to the size of $\mathcal{G}$. Thus, in theory, KGExplainer can balance the computational complexity by adjusting the size of enclosing subgraphs. Besides, KGExplainer can scale to large KGs with a small average degree. More details refer to Appendix A.3.

## 5 EXPERIMENT

In this section, we evaluate the proposed KGExplainer by considering the following *key* research questions: **Q1** Can KGExplainer perform comparable faithfulness and explore informative explanations? **Q2** Can KGExplainer work well with different sizable explainable subgraphs and various KGE methods? **Q3** Is KGExplainer efficient in exploring explanations? We provide more implementation details of KGExplainer and baselines in Appendix B. More additional experiments are shown in Appendix C.

## 5.1 EXPERIMENTAL SETTINGS

**Datasets.** We conduct experiments on three widely used datasets for the KGC task. Specifically, **OGB-biokg** is a large-scale KG from OGB project (Hu et al., 2021), which contains millions of facts and is created from a large number of biomedical data repositories. The **Family-rr** (Sadeghian et al., 2019) is a dataset that contains the bloodline relationships between individuals of various families. **FB15k-237** (Bordes et al., 2013b) is a subset of Freebase (Bollacker et al., 2008), which is a large-scale knowledge graph containing general facts. For every dataset, we split it into training and test subsets for the following evaluations of faithfulness and explainability. The detailed statistics of all datasets are shown in Appendix B.1.1.

**Faithful Metrics.** The evaluation of KGC models is performed by running head and tail predictions on each triple in the test set. For each prediction, the ground truth is ranked against all the other entities $\mathcal{V}$. Without loss of generality, for tail prediction, this corresponds to:

$$tailRank(h, r, t) = |\{v \in \mathcal{V} | \phi(h, r, v) >= \phi(h, r, t)\}|. \tag{9}$$

Following Kadlec et al. (2017), we adopt $Hits@N$ to quantify the predictive performance of KGE and distilled models. $Hits@N$ is the fraction of ranks $S_{rank}$ with value $\leq N$:

$$Hits@N = \frac{|\{s_i \in S_r : s_i \leq N\}|}{|S_r|}, \tag{10}$$

where $s_i$ is the ranking of $i$-th candidate entity and $S_r$ is a set of rankings for all candidate entities. Following the method proposed by Kadlec et al. (2017), we focus on the $Hits@1$ metric to better highlight model differences. Based on the designed metrics, we report the mean and variance for five runs.

**Explainable Metrics.** As the ground truth explanations of KGC tasks are usually missing, it is not easy to quantitatively evaluate the quality of explanations. We consider explanations to be sufficient if they allow preserving the KGC performance (Akrami et al., 2020; Wang et al., 2021; Wu et al., 2023). Thus we define the $Recall@N$ and $F1@N$ as the metrics to evaluate the effectiveness of explored explanations. Specifically, $Recall@N$ measures the fidelity of explanatory subgraphs by feeding them into the distilled model and examining how well they recover the target predictive rankings. Given the rankings of tested facts by forwarding their explanations to the distilled evaluator, we calculate the $Recall@N$ as follows:

$$Recall@N = \frac{|\{s_i \in S_r : s_i \leq N\} \cap \{s_j \in S_r^{exp} : s_j \leq N\}|}{|\{s_i \in S_r : s_i \leq N\}|}, \tag{11}$$

where $S_r^{exp}$ is the ranking set of explanations and $S_r$ denotes the original predictive rankings of them. We can assess the trustworthiness of explanations by calculating the recovery rate of the rankings $S_r$. Meanwhile, the synergistic path-based explanations are critical in the precision of rankings (i.e., $Hits@N$), to comprehensively evaluate the explainability of KGExplainer, we define $F1@N$ to balance the recall and precision of the explored explanations:

$$Hits@N = \frac{|\{s_j \in S_r^{exp} : s_j \leq N\}|}{|S_r^{exp}|}, F1@N = \frac{2 \cdot Recall@N \cdot Hits@N}{Recall@N + Hits@N}. \tag{12}$$

Finally, we consider $N \in \{1, 3, 10\}$ for the qualitative metrics to emphasize the performance discrepancy.

**Baselines.** To verify the performance of KGExplainer in exploring explanations, we compare it against subgraph-based (i.e., a batch of discrete facts) methods **CRIAGE** (Pezeshkpour et al., 2018) and **Kelpie** (Rossi et al., 2022), and path-based **DRUM** (Sadeghian et al., 2019), **ELEP** (Bhowmik & de Melo, 2020), and **PaGE-Link** (Zhang et al., 2023). The sources of subgraph-based models KGEx (Baltatzis & Costabello, 2024) and KE-X (Zhao et al., 2023) are unavailable now, so we exclude them as our baselines.

Table 1: The predictive performance of KGE and the distilled models on $Hits@1(\%)$ metric.

| Methods | OGB-biokg | Family-rr | FB15k-237 |
|---|---|---|---|
| TransE | 11.32 (0.22) | 12.63 (0.09) | 41.86 (0.43) |
| KGExp-TransE | 10.40 (0.25) | 15.77 (0.11) | 40.99 (0.31) |
| DistMult | 16.11 (0.06) | 8.680 (0.27) | 10.26 (0.21) |
| KGExp-DistMult | 14.04 (0.11) | 9.881 (0.14) | 13.09 (0.32) |
| RotatE | 17.12 (0.19) | 39.08 (0.45) | 42.10 (0.51) |
| KGExp-RotatE | 17.33 (0.12) | 35.79 (0.06) | 39.28 (0.29) |

Furthermore, we design variants of KGExplainer: **KGExp-Rand** randomly selected some connective paths between head and tail entities to construct an explainable subgraph, which allows an ablation study. **KGExp-TransE**, **KGExp-DistMult**, and **KGExp-RotatE** are variants of KGExplainer that consider TransE (Bordes et al., 2013b), DistMult (Yang et al., 2015), and RotatE(Sun et al., 2019) as the base KGE, respectively. For convenience, we denote **KGExplainer** as **KGExp**.

## 5.2 Performance of Faithfulness and Explainability

In response to **Q1**, we analyze the performance of KGExplainer from faithfulness and explainability. Besides, we provide a human evaluation on Family-rr dataset.

**Faithfulness Evaluation.** An important goal of explainability models is to accurately and comprehensively represent the local decision-making structure of the target model. Specifically, in the context of KGE models, KGExplainer should strive to replicate their behavior as closely as possible (Zhang et al., 2021). Thus, to answer **Q1**, we compare the predictive performance of KGExplainer and the target KGE models. As shown in Table 1, we observe that KGExplainer achieves nearly the same performance on the KGC task compared with the target KGE models (e.g., the performance of KGExp-TransE and TransE on FB15k-237 dataset). Furthermore, we observe that RotatE overall achieves the best performance for KGC and similarly KGExp-RotatE shows superior predictive performance and the best recovery rate among the distilled models, which brings greater credibility than others, supporting that KGExplainer is faithful to target KGE and is reliable in replicating the target KGE. Thus, we chose the RotatE as the target KGE model and distilled a powerful evaluator based on the subgraph substructure around the target link to assess the explainability. In Table 1, we can see that KGExplainer benefits the base model for some datasets. We observe the same behavior in (Deng & Zhang, 2020; Jiao et al., 2019). From their insights, we think the KGExplainer

Table 2: The $F1@1(\%)$ performance of explanations over various datasets. The **boldface** denotes the highest score and the underline indicates the best baseline.

| Methods | OGB-biokg | Family-rr | FB15k-237 |
|---|---|---|---|
| CRIAGE | 50.03 (0.09) | 20.75 (0.32) | 40.99 (0.21) |
| Kelpie | 49.32 (0.16) | 20.33 (0.23) | 36.64 (0.13) |
| DRUM | 62.56 (0.23) | 21.26 (0.09) | 65.53 (0.19) |
| ELEP | 63.44 (0.21) | 22.75 (0.17) | 70.11 (0.21) |
| PaGE-Link | 68.29 (0.19) | 21.89 (0.13) | 66.12 (0.14) |
| KGExp-Rand | 54.32 (0.43) | 19.16 (0.26) | 55.17 (0.23) |
| KGExp-TransE | 81.97 (0.13) | 30.47 (0.25) | 82.04 (0.33) |
| KGExp-DistMult | 82.33 (0.03) | **30.49** (0.24) | 84.29 (0.17) |
| KGExp-RotatE | **85.21** (0.11) | 29.52 (0.12) | **83.65** (0.14) |

Table 3: The $Recall@1(\%)$ performance of explanations over various datasets. We mark the best score with bold font and mark the best baseline with underline.

| Methods | OGB-biokg | Family-rr | FB15k-237 |
|---|---|---|---|
| CRIAGE | 58.75 (0.07) | 28.57 (0.33) | 68.23 (0.43) |
| Kelpie | 60.32 (0.22) | 27.62 (0.39) | 60.01 (0.27) |
| DRUM | 69.87 (0.14) | 16.20 (0.21) | 87.88 (0.22) |
| ELEP | 76.23 (0.25) | 29.84 (0.18) | 85.58 (0.29) |
| PaGE-Link | 77.69 (0.15) | 27.14 (0.31) | 85.13 (0.09) |
| KGExp-Rand | 54.61 (0.55) | 26.98 (0.49) | 50.37 (0.19) |
| KGExp-TransE | 88.37 (0.21) | **34.61** (0.43) | **92.07** (0.33) |
| KGExp-DistMult | **89.77** (0.11) | 34.60 (0.13) | 91.67 (0.11) |
| KGExp-RotatE | 88.65 (0.08) | 33.02 (0.11) | 91.45 (0.09) |

distilled from KGEs may remove some noise and irrelevant information to the KGC task, making KGExplainer more robust and having better generalization ability, which leads to KGExplainer performing better than base KGEs to some extent. Based on these observations, the effectiveness of the distilled evaluator is demonstrated empirically.

**Explainability Evaluation.** It is essential to assess objectively how good an explanation is for KGExplainer. Unfortunately, similar to other explanation methods, evaluating explanations is made difficult by the impossibility of collecting *ground truth* explanations. To fairly evaluate explanations without ground truth and answer **Q1**, we distill an evaluator and design two metrics $F1@1$ and $Recall@1$ to quantify how trusty an explanation is. As shown in Table 2 and Table 3, we reported the $F1@1$ and $Recall@1$ results compared with baselines on three wide-used datasets ($F1@[3, 10]$ and $Recall@[3, 10]$ are reported in Appendix C.2). Specifically, KGExplainer improves the $F1@1$ and $Recall@1$ by at least $16.92\%$ and $12.08\%$ respectively on the OGB-biokg dataset and achieves $7.74\%$ and $4.77\%$ absolute increase over the best baseline on the Family-rr dataset. Meanwhile, the performance of KGExplainer on the FB15k-237 dataset has yielded $13.54\%$ and $4.19\%$ gains in $F1@1$ and $Recall@1$ compared with the baselines.

Furthermore, we have the following observations: (1) Compared with the subgraph-based methods CRIAGE and Kelpie, the path-based explainability methods DRUM, ELEP, and PaGE-link that

utilize reasonable paths between source and target entities achieve better performance, which indicates that the connecting paths are important for the model explanation. (2) Compared with the path-based explainability models ELEP and PaGE-Link, synergistic path-based KGExplainer (e.g., KGExp-RotatE) performs better than them on all datasets, demonstrating that the multiple synergistic path-based patterns are more effective than paths and multiple disconnected facts.

(3) For an ablation study, KGExp-Rand, which adopts randomly sampled subgraphs connected to the source and target entities as the explanations, yields significantly inferior performance compared with other KGExplainer variants. This demonstrates that KGExplainer indeed identifies semantically meaningful connected subgraph explanations for model predictions, which are superior to randomly sampled subgraphs by a large margin. This may be because the KGExplainer based on target KGEs can explore synergistic reasonable paths that are crucial to predicting the corresponding facts, and the randomly sampled subgraph may contain irrelevant information with missing key patterns to the predicted facts. The overall performance compared with the baselines demonstrates that KGExplainer effectively explores and evaluates informative explanations.

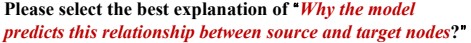

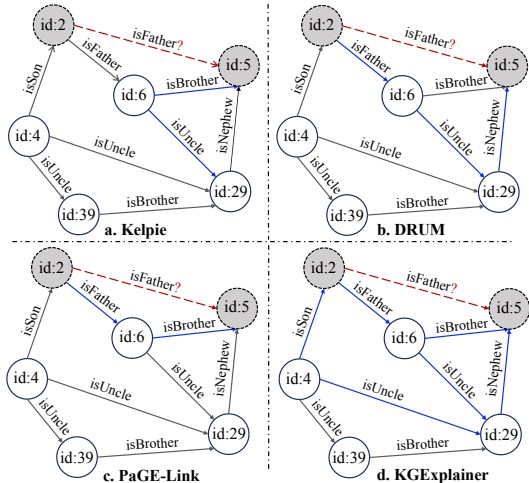

**Human Evaluation** Following Zhang et al. (2023), we conduct a human evaluation by randomly picking 20 predicted links from the test set of Family-rr and generating explanations for each link using a subgraph-based model (i.e., Kelpie), two path-based methods (i.e., DRUM and PaGE-Link), and KGExplainer. We designed a survey with single-choice questions. In each question, we represent the predicted link and those four explanations with both the graph structure and the node/edge type information, similarly as in Figure 3 but excluding method names. We sent the survey to people across graduate students, PhD students, and professors, including people with and without background knowledge about KGs. We ask respondents to "please select the best explanation

Figure 3: The explanations explored by Kelpie, DRUM, PaGE-Link, and KGExplainer on the fact $\langle id : 2, isFather, id : 5 \rangle$.

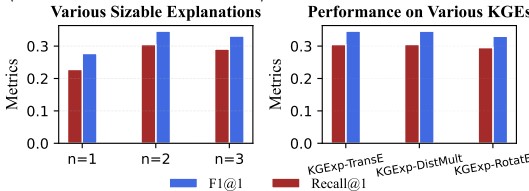

Figure 4: The explainable performance of KG-Explainer with different hyper-parameters over Family-rr dataset.

of ***Why the model predicts this relationship between source and target nodes?***". At least three answers from different people are collected for each question. In total 78 evaluations were collected and 83.3% (65/78) of them selected explanations by KGExplainer as the best. The survey demonstrates that KGExplainer is more effective in searching for human-understandable explanations than other methods. More details refer to Appendix B.2.

## 5.3 ROBUSTNESS OF KGEXPLAINER (Q2)

In this section, to answer **Q2**, we conduct the robustness analysis on the Family-rr dataset to study the impact of different sizable explored subgraphs and various KGE-based KGC models.

**Size of Synergistic Path-based explanations.** To investigate the influence of different sizable path-based explanations, we conduct experiments based on KGExp-RotatE by varying $n$ to *1*, *2*, and *3*. The KGExplainer explored synergistic paths with $n = 1$ is equal to exploring a single explainable path. As illustrated in the left of Figure 4, we observe the best performance on $F1@1$ and $Recall@1$ metrics when utilizing the explainable paths with $n = 2$. Meanwhile, the performance across different sizes collapsed into a hunchback shape. The reason could be that the explored synergistic paths with $n = 1$ are inefficient in recovering the original predictions and the subgraph patterns

with $n = 3$ may contain many irrelevant facts, which introduce much noise and reduce the performance of KGExplainer. This suggests that KGExplainer maintains comparable performance under different sizable synergistic paths.

**KGExplainer over various KGE-based KGC models.** We investigate the impact of different target KGEs, including TransE, DistMult, and RotatE. As shown on the right side of Figure 4, the results indicate that KGExplainer achieves similar levels of explanation performance across $F1@1$ and $Recall@1$ for different variants. This is because KGExplainer selects key entities by perturbing neighboring entities and evaluating their influence on the prediction score, which is independent of the KGE-based KGC models. Our findings show that KGExplainer is effective in searching explanations for embedding-based models and is model-agnostic, as demonstrated by its performance across various KGEs.

## 5.4 EFFICIENCY ANALYSIS (Q3)

**Inference Time.** We trained KGExplainer and baselines using a server with 12 virtual Intel(R) Xeon(R) Platinum 8255C CPU and one RTX 2080 Ti GPU. To verify the effectiveness of KGExplainer in searching for explanations, we designed an experiment to evaluate KGExplainer. As illustrated in Table 4, we show the average cost time of exploring explanations over the test sets of all datasets from different methods. We find that KGExplainer has a similar cost to CRIAGE and is more efficient than PaGE-Link and Kelpie, which indicates that KGExplainer can be effectively extended to complex KGs in searching synergistic path-based explanations. This is because KGExplainer searches explanations within the enclosing subgraph and the number of calculations for each hop is only related to the average degree, which reduces the computational complexity of the exploring process. Overall, KGExplainer has superiority in exploring synergistic path-based explanations with complex patterns, which can scale to large knowledge graphs.

**Sparsity of Explanations**

We discuss the sparsity of the explanations from various explainers. For KGExplainer and baseline methods, we adopt the average number of edges (i.e., **#edge**) as the size of explanations. For KGExplainer, the users can adjust the sparsity of explanation subgraphs by selecting different top-$n$ nodes per hop (Refer to Section 4.2). As shown in Table 5, we observe that the path-based method (e.g., PaGE-Link) exploring a coherent reasoning chain with fewer facts is more efficient than the subgraph-based method (e.g., Keplie). Additionally, KGExplainer exploring multiple synergistic paths can achieve superior performance on comparable sparse explanations than the path-based method. The phenomenon in Table 5 shows that KGExplainer can yield better explainability on explanations with low information density.

Table 4: The average cost time (s) of exploring explanations for various methods. Ours shows comparable efficiency.

| | CRIAGE | Kelpie | PaGE-Link | Ours |
|---|---|---|---|---|
| **Family-rr** | 1.65 | 4.84 | 1.74 | **1.32** |
| **OGB-biokg** | **121.67** | 183.24 | 257.35 | 157.28 |
| **FB15k-237** | **15.13** | 28.51 | 16.93 | 16.81 |

Table 5: The sparsity and corresponding performance of different explainers. Ours indicates KGExplainer shows superior sparsity.

| Methods | Family-rr | | OGB-biokg | | FB15k-237 | |
|---|---|---|---|---|---|---|
| | #edge | F1@1 | #edge | F1@1 | #edge | F1@1 |
| Keplie | 8.297 | 20.33 | 37.58 | 49.32 | 11.19 | 36.64 |
| PaGE-Link | 3.085 | 21.89 | 12.87 | 68.29 | 4.993 | 66.12 |
| Ours ($n = 1$) | 3.182 | 28.77 | 9.751 | 79.28 | 5.142 | 75.67 |
| Ours ($n = 2$) | 9.782 | 30.47 | 45.27 | 81.97 | 13.66 | 82.04 |
| Ours ($n = 3$) | 15.13 | 30.49 | 112.3 | 80.33 | 26.14 | 83.21 |

## 6 CONCLUSION

In this work, we proposed a KGExplainer to identify synergistic paths as explanations over complex KGs and design two quantitative metrics for evaluating them. Specifically, KGExplainer used a greedy search algorithm and distilled an evaluator to find and assess key paths. Extensive experiments demonstrate that KGExplainer is effective in providing meaningful explanations. In the future, we will generalize our framework into multiple domains, including but not limited to AI for chemical and biological applications.

ACKNOWLEDGEMENT

The work was supported by the National Natural Science Foundation of China (U22A2037, 62272151, 62122025, 62425204, 62450002 and 62432011), the National Science and Technology Major Project (2023ZD0120902), Hunan Provincial Natural Science Foundation of China (2022JJ20016), The science and technology innovation Program of Hunan Province (2022RC1099), and the Beijing Natural Science Foundation (L248013).

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

---

**Algorithm 1:** Greedy search for subgraph explanation

---

**Input** : Target prediction $\langle h, r, t \rangle$; Enclosing subgraph $g$; Prediction score $s$; Subgraph
              extracting function $F_{sub}$; Score function $\phi$; Number of entities retrieved per hop $n$;
**Output:** A key subgraph $g_{key}$ that is most critical to the target prediction $\langle h, r, t \rangle$.
Initialize a set of visited entities $S_{visit}$ as $\{h\}$;
Initialize a set of key entities $S_{key}$ as $\{h\}$;
Initialize a queue of searching entities $S_{search}$ as $\{h\}$;
**while** $S_{search}$ *is not empty* **do**
    Initialize $H$ as an empty max-Heap;
    $e_{searching} \leftarrow S_{search}.pop()$;
    **if** $t \in \mathcal{N}(e_{searching})$ **then**
        $S_{key} \leftarrow S_{key} \cup \{t\}$ ;
        break;
    **for** $v \in \mathcal{N}(e_{searching})$ **do**
        **if** $u \notin S_{visit}$ **then**
            $S_{visit} \leftarrow S_{visit} \cup \{u\}$ ;
            Create $g'$ by removing $u$ from $g$ and finetune the KG embeddings of $g'$;
            Get new embeddings of $h$, $r$, $t$ as $\mathbf{h'}$, $\mathbf{r'}$, $\mathbf{t'}$;
            $s' \leftarrow \phi(\mathbf{h'}, \mathbf{r'}, \mathbf{t'})$;
            $\delta_v \leftarrow s - s'$;
            Put $v$ into $H$ with its corresponding value $\delta_v$;
    **for** $i \leftarrow 1$ *to* $n$ **do**
        Pop $v$ with the maximum $\delta_v$ from $H$ and put it into $S_{key}$ and $S_{search}$;
$g_{key} \leftarrow F_{sub}(g, S_{key})$;
**return** $g_{key}$;

---

Table 6: The statistics of datasets.

| Dataset | #entity | #relation | #train | #test |
|---------|---------|-----------|--------|-------|
| **WN-18** | 40,943 | 18 | 35,354 | 2,250 |
| **Family-rr** | 3,007 | 12 | 5,868 | 2,835 |
| **FB15k-237** | 14,541 | 237 | 68,028 | 9,209 |
| **OGB-biokg** | 93,773 | 51 | 4,762,678 | 3000 |

# A KGEXPLAINER

## A.1 THE ALGORITHM OF GREEDY SEARCH

For easier understanding, we show a detailed algorithm to describe how KGExplainer provides key explanations in Algorithm 1.

## A.2 RETRAINING STRATEGY

Retraing the embeddings over the entire KG $\mathcal{G}$ is infeasible due to removing one entity and its facts have little impact on the embedding of KG elements. To reduce the overhead of retraining, we developed a subgraph-based retraining strategy as shown in Figure 5. We extract a 2-hop enclosing subgraph $g$ around the target fact $\langle h, r, t \rangle$. We argue that the subgraph $g$ contains the key information that leads to the target prediction. Thus the retraining process is only visible to the subgraph. Figure 5 shows an example of how $g$ is extracted. Meanwhile, to ensure that the retrained entity representation space is consistent with the original knowledge graph, we expand $g$ by including the 1-hop neighbors $\mathcal{N}_{extend}$ of the existing entities and fixed their embeddings. In Figure 5, we show the learnable and fixed entities in the fine-tuning process. Specifically, we adopt the following constraints: (a) we use the embeddings of the entities in $\mathcal{G}$ to initialize the embeddings of $g$; (b) we fixed the relation embeddings in $g$ as the same as those in $\mathcal{G}$; and (c) we fixed the embedding of $v \in \mathcal{N}_{extend}$.

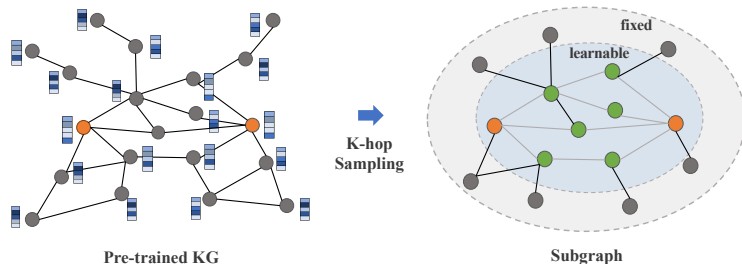

Figure 5: An example of creating a subgraph from $\mathcal{G}$. On the left, two entities (the orange vertices) were predicted to be connected. A subgraph was created on the right. We first initialized $g$ with all the entities (the orange and green vertices) within $\mathcal{G}$. Then it was expanded by including the 1-hop neighbors $\mathcal{N}_{extend}$ of existing entities (the gray vertices).

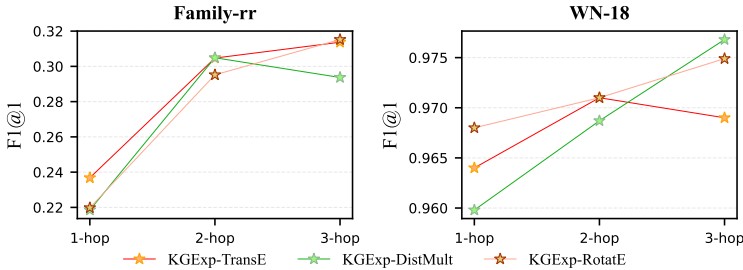

Figure 6: The $F1@1$ performance of KGExplainer on Family-rr and WN-18 over various sizable enclosing subgraphs.

## A.3 THE COMPUTATIONAL EFFICIENCY

The computational complexity of KGExplainer primarily arises from two factors: (1) the greedy search within $g$, and (2) the re-training cost associated with $g$. As discussed in Section 4.4, the complexity of the greedy search is largely determined by local parameters. The re-training cost, on the other hand, depends on the underlying KGE model The computational complexity of the re-training stage for different base KGEs, which is dominated by the size of subgraph $g$ (i.e., $O(\frac{LN|\mathcal{V}|}{|\mathcal{E}|})$ discussed in Section 4.4), the embedding dimension $d$, and the size of negative samples $n$. Thus we can analyze the efficiency of base KGEs as follows:

- In TransE, DistMult, and RotatE, the computational efficiency of each fact within $g$ is $O((n+1)d)$. Therefore, the overall complexity is $O(\frac{LN|\mathcal{V}|}{|\mathcal{E}|}(n+1)d)$, where $L$ is the maximum path length, $N$ denotes the path number.

In conclusion, the practical runtime largely depends on the local parameters of the knowledge graphs. The results in Table 7 show a strong positive correlation between the number of paths and computation time, which aligns closely with our theoretical analysis. Therefore, we can improve the scalability of large knowledge graphs by adjusting these local parameters.

Table 7: The distribution of degree, path length, and path number over various KGs. Note that the analysis is conducted within local subgraphs (2-hop) of test set.

| Datasets | avg. degree | avg. path length | avg. path number | avg. time |
|----------|-------------|------------------|------------------|-----------|
| Family-rr | 3.902 | 2.828 | 4.1 | 1.32 |
| FB15k-237 | 9.357 | 4.195 | 50.054 | 16.81 |
| OGB-biokg | 101.579 | 3.982 | 457.234 | 157.28 |

A.4 Formalizing the Concept of Path Synergy

We can formalize the concept of synergistic paths from an information-theoretic perspective.

Let:

- $(G = (V, R, E))$ be a KG with entities $V$, relations $R$, and edges $E$.
- $< h, r, t >$ denote the head entity, relation, and tail entity, respectively, forming the target prediction.
- $\mathcal{P} = \{p_1, p_2, \ldots, p_n\}$ be a set of paths between $h$ and $t$, where each $p_i$ represents a sequence of edges from $h$ to $t$.

We aim to quantify the *synergy* among paths in $\mathcal{P}$ by evaluating how much additional information these paths provide collectively toward the prediction $t$, compared to the information provided by each path individually.

1. **Mutual Information of Individual Paths**: Define $I(t; p_i)$ as the mutual information between the target prediction $t$ and a path $p_i$ from $h$ to $t$. This quantifies the amount of information that $p_i$ alone contributes to the prediction of $t$:

$$I(t; p_i) = H(t) - H(t|p_i),$$

where $H(t)$ is the entropy of the target prediction $t$, and $H(t|p_i)$ is the conditional entropy of $t$ given the information from path $p_i$.

2. **Joint Mutual Information of Path Set**: Let $I(t; \mathcal{P}_{synergy})$ represent the mutual information between $t$ and the subset of paths $\mathcal{P}_{synergy} \subset \mathcal{P}$, which includes multiple paths that contribute to the prediction. This joint mutual information measures the information provided by the paths in $\mathcal{P}_{synergy}$ when considered together:

$$I(t; \mathcal{P}_{synergy}) = H(t) - H(t|\mathcal{P}_{synergy}).$$

3. **Synergy Condition**: The paths in $\mathcal{P}_{synergy}$ are considered *synergistic* if the joint mutual information $I(t; \mathcal{P}_{synergy})$ is greater than the sum of individual mutual informations $I(t; p_i)$ for each $p_i \in \mathcal{P}_{synergy}$:

$$I(t; \mathcal{P}_{synergy}) > \sum_{p_i \in \mathcal{P}_{synergy}} I(t; p_i).$$

This inequality implies that the paths in $\mathcal{P}_{synergy}$ provide *additional predictive information* about $t$ when considered together, beyond what each path contributes individually. This excess information captures the synergy among paths.

4. **Interpretation and Optimization**:

- The set $\mathcal{P}_{synergy}$ can be identified as the subset of paths $\mathcal{P}$ from $h$ to $t$ that maximizes the *synergistic mutual information* $I(t; \mathcal{P}_{synergy}) - \sum_{p_i \in \mathcal{P}_{synergy}} I(t; p_i)$.
- Formally, we can define the optimal synergistic path set $\mathcal{P}^*_{synergy}$ as:

$$\mathcal{P}^*_{synergy} = \arg\max_{\mathcal{P}' \subseteq \mathcal{P}} \left( I(t; \mathcal{P}') - \sum_{p_i \in \mathcal{P}'} I(t; p_i) \right).$$

# B Experimental Settings

## B.1 Implementation Details

### B.1.1 Datasets

We show detailed statistics of all datasets in Table 6. #train and #test are the number of triples on the training and testing sets.

Table 8: The hyperparameters of KGExplainer during distillation phase.

| | **Family-rr** | **WN-18** | **FB15k-237** | **OGB-biokg** |
|---|---|---|---|---|
| embedding dim | 64 | 64 | 64 | 64 |
| learning rate | 0.0015 | 0.001 | 0.001 | 0.001 |
| weight of pair-wise loss | 0.5 | 0.5 | 0.5 | 0.5 |
| Epochs | 50 | 50 | 50 | 100 |
| R-GAT layers | 2 | 3 | 3 | 3 |
| Optimizer | Adam | Adam | Adam | Adam |
| Batch size | 1024 | 4096 | 4096 | 4096 |
| #entity per hop $n$ | 2 | 2 | 2 | 2 |
| enclosing subgraph size $k$ | 2 | 2 | 1 | 2 |
| target KGE | TransE | RotatE | RotatE | RotatE |

### B.1.2 DETAILS OF KGEXPLAINER

The experiments contain (1) Generating subgraph-based explanations, and (2) Distilling the subgraph evaluator for assessment.

In the *explanation phase*, we extract key subgraphs from the 2-hop enclosing subgraph surrounding the target prediction. We set $n = 2$ to determine the number of selected entities per hop during greedy searching. To achieve the best evaluation capability, we distill an evaluator from RotatE Sun et al. (2019) to assess explainability (See Section 5.2). More settings of hyperparameters are shown in Table 8. In the *distilling phase*, to ensure sufficient expressive power for the model to distill, we tune the hyperparameters to get the optimal distilled model. The details of hyperparameters are depicted in Table 8.

### B.1.3 DETAILS OF BASELINES

We implement our KGExplainer and baseline models in Pytorch Paszke et al. (2017) and DGL Wang et al. (2019b) library on an RTX 2080Ti GPU with 11GB memory. For the implementation of baselines, we use the source code of Kelpie Rossi et al. (2022) to reproduce the faithful results and explore explanations of fact-based methods CRIAGE Pezeshkpour et al. (2018) and Kelpie. For other baselines, we adopt the source code reported by their original paper and use their pre-defined parameters to produce explanations. In addition to comparing against baselines with no parameter settings for certain datasets, we also examine the parameters used in a similar scale dataset. We report the average results and their standard deviation by five runs in Section 5.

### B.2 DETAILS OF HUMAN EVALUATION

For valuable human evaluations, we consider the following guidance for participants:

- We give two bases for judgment as a guide when asking participants questions. 1. In Family-rr, for predicting the kinship between two individuals, the explanation should be based on the known family memberships, marital relationships, etc. reasonably derived. 2. Encourage testers to choose more concise and clear explanations, while maintaining completeness. Avoid overly complex explanations so that key information can be presented clearly.

- The results from different models are presented to the participants simultaneously

- We randomly selected 20 samples from the test set of Family-rr for participants to evaluate.

## C ADDITIONAL EXPERIMENTS

### C.1 PERFORMANCE ON VARIOUS SIZABLE ENCLOSING SUBGRAPHS

Similar to the experiments in Section 5.3 (i.e., *Hyper-parameter Sensitivity Analysis*), we conduct another experiment to study the model's explainability on various sizable enclosing subgraphs. We

Table 9: The $F1@N(\%)$ performance of explanations over various datasets. The boldface denotes the highest score.

| | Family-rr | | WN-18 | | FB15k-237 | | OGB-biokg | |
| | F1@3 | F1@10 | F1@3 | F1@10 | F1@3 | F1@10 | F1@3 | F1@10 |
|---|---|---|---|---|---|---|---|---|
| CRIAGE | 27.22 | 36.44 | 72.91 | 77.78 | 48.81 | 53.26 | 54.27 | 61.62 |
| Keplie | 27.18 | 36.25 | 73.58 | 79.39 | 40.77 | 48.01 | 52.59 | 61.21 |
| DRUM | 25.23 | 30.73 | 75.58 | 77.80 | 67.34 | 71.03 | 65.87 | 72.54 |
| ELPE | 75.69 | 79.27 | 78.35 | 83.21 | 29.71 | 40.40 | 67.41 | 71.65 |
| PaGE-Link | 28.42 | 38.49 | 70.99 | 75.21 | 70.77 | 76.51 | 71.32 | 77.58 |
| KGExp-rand | 27.55 | 41.48 | 37.43 | 40.46 | 61.03 | 66.58 | 56.47 | 59.39 |
| KGExp-TransE | 45.89 | 62.96 | 98.57 | 99.58 | 85.41 | 89.97 | 84.22 | 88.37 |
| KGExp-DistMult | 44.08 | 57.47 | 97.89 | 99.41 | 86.74 | 90.17 | 84.35 | 89.73 |
| KGExp-RotatE | 39.56 | 57.25 | 98.58 | 99.57 | 87.41 | 92.97 | 86.97 | 90.04 |

Table 10: The $Recall@N(\%)$ performance of explanations over various datasets. The boldface denotes the highest score.

| | Family-rr | | WN-18 | | FB15k-237 | | OGB-biokg | |
| | Recall@3 | Recall@10 | Recall@3 | Recall@10 | Recall@3 | Recall@10 | Recall@3 | Recall@10 |
|---|---|---|---|---|---|---|---|---|
| CRIAGE | 32.02 | 36.95 | 95.45 | 99.34 | 76.87 | 82.11 | 65.95 | 70.44 |
| Keplie | 31.87 | 36.55 | 93.69 | 99.51 | 66.91 | 75.52 | 67.26 | 75.34 |
| DRUM | 22.09 | 30.92 | 93.69 | 94.93 | 89.01 | 91.12 | 76.23 | 81.07 |
| ELPE | 35.17 | 41.34 | 93.11 | 95.23 | 89.76 | 93.21 | 80.21 | 83.15 |
| PaGE-Link | 31.77 | 38.13 | 88.32 | 92.20 | 87.32 | 89.04 | 78.65 | 83.26 |
| KGExp-rand | 33.44 | 43.97 | 92.47 | 94.61 | 57.12 | 64.31 | 58.37 | 62.59 |
| KGExp-TransE | 46.85 | 61.61 | 99.47 | 99.84 | 93.41 | 94.57 | 90.48 | 93.51 |
| KGExp-DistMult | 43.05 | 56.26 | 99.31 | 99.71 | 92.01 | 93.89 | 91.66 | 93.05 |
| KGExp-RotatE | 41.48 | 56.26 | 99.47 | 99.84 | 93.01 | 94.69 | 90.89 | 92.31 |

select top-$n = 2$ entities per hop and vary the size of $k$-hop enclosing subgraph to $k = 1, 2, 3$. The $F1@1$ performance of KGExplainer over various sizable subgraphs is depicted in Figure 6. Theoretically, the larger the subgraph to be searched, the more sufficient key information it contains, so that more effective explanatory subgraphs can be retrieved. We can observe that KGExp-RotatE performs better as the size of subgraphs increases on both Family-rr and WN-18 datasets, which indicates that KGExplainer based on RotatE has generalization and stability. However, KGExplainer over TransE and DistMult only fits this phenomenon on one of the datasets, showing they are limited in generalizing to other data distributions. Thus to achieve a fair comparison, we consider the KGExp-RotatE as the subgraph evaluator and assess other baseline methods.

## C.2  THE FULL RESULTS OF EXPLANATIONS

The full results (i.e., $F1@[3, 10]$ and $Recall@[3, 10]$) of the explanations are shown in Table 9 and Table 10. We can observe that KGExplainer is also the best.

## C.3  SPARISITY OF EXPLAINERS

We discuss the sparsity of the explanations from various explainers. For subgraph- and path-based methods, we directly adopt the edges and paths for further evaluation. Therefore the size of explanations depends on the number of edges and path lengths they explore. For KGExplainer, the users can adjust the sparsity of multiple synergistic paths by selecting different top-$n$ nodes per hop (See Section 4.4.1). Once the model is trained, the user can directly adjust the value of $n$ without retraining. For various explainers, We present the average number of edges (i.e., **Avg. #edges**) and corresponding $F1@1$ performance on all datasets in Table 11. We see that the path-based method (e.g., PaGE-Link) with a complete reasoning chain is more effective than the fact-based method (e.g., Keplie) on fewer visible facts. Moreover, we observe that KGExplainer can dynamically ad-

Table 11: The sparsity and corresponding performance of different explainers.

| Methods | Family-rr | | WN-18 | | FB15k-237 | | OGB-biokg | |
|---|---|---|---|---|---|---|---|---|
| | Avg. #edges | F1@1 | Avg. #edges | F1@1 | Avg. #edges | F1@1 | Avg. #edges | F1@1 |
| Keplie | 8.297 | 20.33 | 4.794 | 66.37 | 11.19 | 36.64 | 37.58 | 49.32 |
| Page-Link | 3.085 | 21.89 | 3.64 | 67.7 | 4.993 | 66.12 | 12.87 | 68.29 |
| KGExp-TransE (n=1) | 3.182 | 28.77 | 2.263 | 96.4 | 5.142 | 75.67 | 9.75 | 79.28 |
| KGExp-TransE (n=2) | 9.782 | 30.47 | 4.772 | 97.1 | 13.661 | 82.04 | 45.27 | 81.97 |
| KGExp-TransE (n=3) | 15.132 | 30.49 | 9.223 | 96.91 | 26.146 | 83.21 | 112.37 | 80.33 |

just the size of the explanations and achieve superior performance compared with subgraph- and path-based methods.

## C.4 Performance of KGExplainer on Various Architectures

The evaluators built on different architectures demonstrate robustness. The evaluation results of evaluators distilled from different base KGEs have minor differences. Specifically, we evaluate KGExplainer's performance using different evaluator architectures across various KGEs. We use RotatE as the base KGE to generate explanations for the Family-rr dataset, and then assess these explanations using evaluators distilled from different teacher KGEs. Table 12 reveals two key findings:

- The evaluator distilled from RotatE achieves the best performance, with differences among evaluators due to variations in their predictive abilities.

- Evaluators based on R-GAT and GraiL Teru et al. (2020) (which applies R-GCN on subgraphs) show similar performance across different KGEs. This may be because that the architectural differences between R-GAT and R-GCN are minor.

These results highlight that an evaluator with strong predictive capability is essential for high-quality evaluations. Accordingly, we primarily select RotatE as the teacher KGE for distilling evaluators.

Table 12: The explainable performance of various evaluators on the Family-rr dataset.

| teacher KGEs | KGExplainer (R-GAT) | KGExplainer (GraiL) |
|---|---|---|
| TransE | 27.39 | 28.51 |
| DistMult | 23.43 | 21.15 |
| RotatE | 30.47 | 28.26 |

## C.5 Computational Time of KGExplainer on Different Local Parameters

To improve scalability for larger KGs, we can adjust the local enclosing subgraph size (e.g., reducing from a 2-hop to a 1-hop subgraph) to decrease path length and path count. Table 13 presents the distribution of node degree, path length, path number, and computation time across different subgraph scales over OGB-biokg, revealing a strong positive correlation between them, consistent with our theoretical analysis. In conclusion, the complexity of KGExplainer is driven primarily by local parameters—such as path length $L$, path number $N$, and average degree—rather than the overall size of the knowledge graph (KG).

Table 13: The distribution of average degree, path length, path number, and computation time over various subgraph scales.

| Scale | avg. degree | avg. path_length | avg. path number | avg. time |
|---|---|---|---|---|
| 1-hop | 97.265 | 2.573 | 346.331 | 103.245 |
| 2-hop | 101.579 | 3.982 | 457.234 | 157.28 |

# D    DISCUSSION

## D.1    NOVELTY

Some related works Han et al. (2020); Joshi & Urbani (2020) mainly focus on pruning subgraphs to eliminate irrelevant facts and to improve the performance of link prediction on either temporal or static KGs. To some extent, the pruned subgraphs can be considered key information for predicted links. However, the pruned subgraphs are not connected but rather a collection of facts, which are fact-based explainable methods. Similarly, KE-X Zhao et al. (2023) and KGEx Baltatzis & Costabello (2024) are post-hoc explainable methods that identify key triples as explanations. The identified key triples are discrete facts and can not form a reasoning chain. Additionally, PaGE-Link Zhang et al. (2023) and Power-Link Chang et al. (2024) are path-based explainable methods that focus on exploring a single path and rely on selecting multiple discrete paths manually for complex scenes. In contrast, KGExplainer emphasizes exploring multiple synergistic paths globally to form connected subgraphs, which is novel and more expressive than previous models.

## D.2    DISCUSSION OF DISTILLED RESULTS

In Table 1, we can see that KGExplainer benefits the base model for some datasets. We observe the same behavior in Deng & Zhang (2020); Jiao et al. (2019). From their insights, we think the KGExplainer distilled from KGEs may remove some noise and irrelevant information to the KGC task, making KGExplainer more robust and having better generalization ability, which leads to KGExplainer performing better than base KGEs to some extent.

