# OpenReview forum: "Towards Synergistic Path-based Explanations for Knowledge Graph Completion: Exploration and Evaluation"
_ICLR.cc/2025/Conference — ICLR 2025 Poster_

### Official Review · Reviewer_c8rP · 2024-11-01

**Soundness:** 3
**Presentation:** 3
**Contribution:** 3
**Rating:** 6
**Confidence:** 4

**Summary:**

The paper introduces KGExplainer, a novel model-agnostic framework for generating explainable knowledge graph completion (KGC) by identifying synergistic path-based explanations. This approach leverages a perturbation-based greedy search to identify influential paths and an evaluator distilled from the target knowledge graph embedding (KGE) model to assess the fidelity of explanations. KGExplainer is evaluated against state-of-the-art explainability baselines, demonstrating significant improvements in both faithfulness and explainability metrics. Experimental results on multiple benchmark datasets reveal KGExplainer's strong human-interpretability and competitive performance in preserving predictive accuracy.

**Strengths:**

1. KGExplainer introduces a model-agnostic approach to generate explainable paths for KGE-based models, which can be applied to various KGC models and datasets.
2. The paper proposes a distilled evaluator to quantitatively assess explanation fidelity, ensuring that the explanations are not only accurate but also reflective of the underlying KGE model's decision-making process.
3. The framework is evaluated on various KGC models and datasets, showing competitive results across different setups and substantial improvements in both F1@1 and Recall@1 metrics, particularly outperforming other path-based methods.

**Weaknesses:**

1. Limited scopes: The proposed method only focuses on the KGE-based methods, while many advanced KGC methods are based on graph neural networks (GNNs) [1] or other deep learning architectures. It would be beneficial to extend the evaluation to these models to provide a more comprehensive comparison. Besides, it only focuses on path-based explanations, while other high-order structures like subgraph patterns or graph motifs could also be considered.

2. Efficiency concerns: The proposed method requires retraining the KGE model to identify the key paths, which could be computationally expensive. Even though the greedy search is efficient, the overall complexity of the method might hinder its scalability to large-scale knowledge graphs.

3. Evaluation metrics: The paper proposes a novel evaluator to assess the fidelity of explanations. However, the quality of the evaluator is not dissected in detail, making it questionable whether the evaluator can accurately capture the KGE model's decision-making process. A more in-depth analysis of the evaluator's performance and its robustness to different KGE models would strengthen the paper's claims.

4. Some experiment settings are not clearly described.

[1] Galkin, M., Yuan, X., Mostafa, H., Tang, J., & Zhu, Z. Towards Foundation Models for Knowledge Graph Reasoning. In The Twelfth International Conference on Learning Representations.

**Questions:**

1. How to assess the quality of the proposed evaluator? Can you provide some experiments to demonstrate its effectiveness in evaluating the fidelity of explanations?

Moreover, how does the architectures of the evaluator, like other subgraph-based method: Grail [1], affect the performance and robustness of evaluation? Will the evaluation results be consistent across different evaluators?

Last, what is the input to the evaluator? Is it purely the graph structures or the embeddings are initialized with the pre-trained KGE models?

2. How the re-training stage of the KGE model affects the overall efficiency of the proposed method? It should be discussed in the complexity analysis section and also included in the cost time analysis of Table 4.

3. In experiments, how is the predictive performance of the "distilled model" evaluated? Is it predicted by the embeddings from re-trained KGE model or inputting subgraphs into the evaluator?

4. How does the hop of enclosed subgraphs affect the performance of the proposed method? The author only set the hop to 2. However, in FB15k-237, the average length of the shortest path connecting head and tail is around 3, which means the hop of 2 might not be enough to capture the long-range dependencies.

[1] Teru, K., Denis, E., & Hamilton, W. (2020, November). Inductive relation prediction by subgraph reasoning. In International Conference on Machine Learning (pp. 9448-9457). PMLR.

---

> ### Author Response · Authors · 2024-11-18
> **A Kind Reply to Dear Reviewer c8rP**
>
> Thank you for your thoughtful and encouraging review. The point-to-point answer is provided below.
>
> **Q1.** Provide more experiments to assess the quality of the proposed evaluator.
>
> **Response:** We evaluate the quality of the proposed evaluator from two perspectives: (1) the fidelity of its predictive performance for potential facts, and (2) its recovery effectiveness compared to base KGEs. As shown in Table 1, KGExplainer demonstrates comparable predictive accuracy and strong recovery performance relative to target KGEs. These results provide experimental evidence that KGExplainer effectively captures the decision-making process of the KGE model.
> |Methods|Family-rr (recovery rate)|OGB-biokg (recovery rate)|FB15k-237 (recovery rate)|
> |-|-|-|-|
> |TransE|12.63|11.32| 41.86|
> |KGExp-TransE|15.77 (93.4)| 10.40 (90.01)|40.99 (89.9)|
> |DistMult|8.68|16.11|10.26|
> |KGExp-DistMult|9.881 (96.7)|14.04 (85.79)|13.09 (95.3)|
> |RotatE|39.08|17.12|42.1|
> |KGExp-RotatE|35.79 (95.3)|17.33 (95.43)|39.28 (98.7)|
> |
>
> Table 1. The predictive performance and recovery rate of the distilled evaluator.
>
> **Q2.** How does the architectures of the evaluator affect the performance and robustness of evaluation? Will the evaluation results be consistent across different evaluators?
>
> **Response:** The evaluators built on different architectures demonstrate robustness. And the evaluation results of evaluators distilled from different base KGEs exist minor differences. Specifically, we evaluate KGExplainer's performance using different evaluator architectures across various KGEs. We use RotatE as the base KGE to generate explanations for the Family-rr dataset, and then assess these explanations using evaluators distilled from different teacher KGEs. Table 2 reveals two key findings:
> 1. The evaluator distilled from RotatE achieves the best performance, with differences among evaluators due to variations in their predictive abilities.
> 2. Evaluators based on R-GAT and GraiL (which applies R-GCN on subgraphs) show similar performance across different KGEs. This may be because that the architectural differences between R-GAT and R-GCN are minor.
>
> These results highlight that an evaluator with strong predictive capability is essential for high-quality evaluations. Accordingly, we primarily select RotatE as the teacher KGE for distilling evaluators.
>
> |teacher KGEs|KGExplainer (R-GAT)|KGExplainer (GraiL)|
> |-|-|-|
> |TransE|27.39|28.51|
> |DistMult|23.43|21.15|
> |RotatE|30.47|28.26|
> |
>
> Table 2. The explainable performance of various evaluators on the Family-rr dataset.
>
> **Q3.** How is the predictive performance of the "distilled model" evaluated? What is the input to the evaluator?
>
> **Response:**
> 1) To assess the evaluator’s predictive performance, we use tail entity prediction for queries of the form <$h, r, ?$>. Specifically, for each candidate entity $c$, we input the local structure surrounding the query <$h, r, c$> into the evaluator to obtain a score. This allows us to rank the correct tail entity and compute the hit-rate performance across test samples.
> 2) During both the distillation and evaluation phases, we rely solely on local graph structures with semantic relations as input for the evaluator to avoid potential data leakage from embeddings pre-initialized with KGE models.

---

> > ### Author Response · Authors · 2024-11-18
> > **A Kind Reply to Dear Reviewer c8rP (Part 2)**
> >
> > **Q4.** How the re-training stage of the KGE model affect the overall efficiency of the proposed method? It should be discussed in the complexity analysis section and also included in the cost time analysis of *Table 4* (in the main paper).
> >
> > **Response:** The efficiency of the re-training stage is dominated by the local parameters (e.g., the size of the subgraph). The cost time shown in *Table 4* (main paper) includes the **total running time** of the greed search and re-training stages. Specifically, the computational complexity of the re-training stage for different base KGEs is dominated by the size of subgraph $g$ (i.e., $O(\frac{LN|\mathcal{V}|}{|\mathcal{E}|})$ discussed in Section 4.4), the embedding dimension $d$, and the size of negative samples $n$. Thus we can analyze the efficiency of base KGEs as follows:
> > + In TransE, DistMult, and RotatE, the computational efficiency of each fact within $g$ is $O((n+1)\*d)$. Therefore, the overall complexity is $O(\frac{LN|\mathcal{V}|}{|\mathcal{E}|}\*(n+1)\*d)$, where $L$ is the maximum path length, $N$ denotes the path number.
> >
> > In conclusion, the practical runtime largely depends on the local parameters of the knowledge graphs. The results in Table 3 show a strong positive correlation between the number of paths and computation time, which aligns closely with our theoretical analysis.
> > |Datasets|avg. degree|avg. path length|avg. path number|avg. time|
> > |-|-|-|-|-|
> > |Family-rr|3.902|2.828|4.1|1.32|
> > |FB15k-237|9.357|4.195 |50.054 |16.81|
> > |OGB-biokg|101.579|3.982 |457.234|157.28|
> > |
> >
> > Table 3. The distribution of degree, path length, path number, and computation time over various KGs.
> >
> > **Q5.** How does the hop of enclosed subgraphs affect the performance of the proposed method?
> >
> > **Response:** In Appendix C.1, we examine the impact of subgraph size on explainability evaluation. Intuitively, larger subgraphs contain more essential information, allowing for the retrieval of more effective explanatory subgraphs. As shown in Table 4, KGExp-RotatE consistently improves as subgraph size increases across the Family-rr, OGB-biokg, and FB15k-237 datasets. The performance improvement of KGExplainer is not significant from 2-hop to 3-hop but imposes more computational burden. Therefore, we adopt 2-hop as our standard configuration.
> > |KGExp-RotatE|Family-rr|OGB-biokg|FB15k-237|
> > |-|-|-|-|
> > |1-hop|0.213|0.815|0.803|
> > |2-hop|0.306|0.852|0.836|
> > |3-hop|0.315|0.834|0.837|
> > |
> >
> > Table 4. The performance of KGExp-RotatE on various datasets for different subgraph scales.
> >
> > Thank you once again for your valuable feedback, which has helped us improve our work. We believe these additions will strengthen our model's presentation.

---

> > > ### Comment · Reviewer_c8rP · 2024-11-18
> > >
> > > Thanks to the author for the response, which addresses most of my questions. But I have some follow-up questions.
> > > 1. How do we calculate the recovery rate for the proposed evaluator? As shown in Table 1 of the response, the evaluator sometimes achieves better performance than the KGE method.
> > > 2. I am still unclear about the process of the evaluator. The author claims that the evaluator only takes subgraph structure as input without the need for KGE. However, the R-GAT adopted as one of the evaluators requires node features as input. In addition, equation 4 in the paper also involves node embedding x_i and relation embedding e_r. How are they initialized?
> > > 3. I have the following concerns about the novelty of developing an explanation method for KGE. Since the KGE methods have been largely beaten by GNN-based methods in KGC tasks (the evaluator also follows the GNN-based architecture), do we still need to focus on explaining KGE? Existing works focus on the explanations for GNN-based KGC methods [1], which also provide good explanations.
> > >
> > > [1] Chang, H., Ye, J., Lopez-Avila, A., Du, J., & Li, J. (2024, August). Path-based Explanation for Knowledge Graph Completion. In Proceedings of the 30th ACM SIGKDD Conference on Knowledge Discovery and Data Mining (pp. 231-242).

---

> > > > ### Author Response · Authors · 2024-11-20
> > > > **A Kind Reply to Dear Reviewer c8rP**
> > > >
> > > > Thank you for your constructive comments. The point-to-point answer is provided below.
> > > >
> > > > **Q1:** How do we calculate the recovery rate for the proposed evaluator?
> > > >
> > > > **Response:** The recovery rate measures how well the evaluator replicates the predictions of the base KGE model. Specifically, it is defined as the percentage of test samples for which the evaluator and the base KGE make the same prediction. The calculation is as follows:
> > > > $$recovery\\\_rate = \frac{\sum_{i=1}^{N}F(\phi, h_i, r_i,t_i)\oplus F(Z, h_i, r_i,t_i)}{N},$$
> > > > $$F(f, h, r, t)=
> > > > \begin{cases}
> > > > 1&\operatorname*{if argmax}_{v\in\mathcal{V}}f(h,r,v)=t\\\\
> > > > 0 & \operatorname*{others}
> > > > \end{cases}$$
> > > > Here, $\phi$ denotes the KGE model, $Z$ denotes the evaluator, $N$ is the number of test samples, $\mathcal{V}$ is the set of possible entities, and $\oplus$ indicates whether $\phi$ and $Z$ make the same prediction. This heterodyne operation ensures that the recovery rate reflects the fidelity of the evaluator to the decision-making process of the base KGE.
> > > >
> > > > **Q2:** I am still unclear about the process of the evaluator. The author claims that the evaluator only takes subgraph structure as input without the need for KGE. However, the R-GAT adopted as one of the evaluators requires node features as input. In addition, equation 4 in the paper also involves node embedding x_i and relation embedding e_r. How are they initialized?
> > > >
> > > > **Response:** We adopt Xavier initializer [1] to initialize the entity and relation embeddings, which avoid potential data leakage during distillation phase for base KGE. We will clarify it in the revised version to avoid further confusion.
> > > >
> > > > **Q3:** I have the following concerns about the novelty of developing an explanation method for KGE. Since the KGE methods have been largely beaten by GNN-based methods in KGC tasks (the evaluator also follows the GNN-based architecture), do we still need to focus on explaining KGE? Existing works focus on the explanations for GNN-based KGC methods [2], which also provide good explanations.
> > > >
> > > > **Response:**
> > > > We understand your concerns. In recent studies, we found that KGE models in KGC have similar performance to GNN [3]. Besides, KGE models remain competitive in specific, high-stakes applications like drug repurposing [4] and drug–target interaction prediction [5], where interpretability and scalability are critical. These scenarios highlight the need to explain KGE methods.
> > > >
> > > > Our KGExplainer is model-agnostic and applicable to both KGE and GNN-based methods. For instance, it can use greedy search and re-training strategies to analyze key patterns in pretrained GNNs like R-GAT, with the fidelity of explanations evaluated using the pretrained GNN. While prior works like [2] have made significant contributions to explaining GNNs, interpretability for KGE remains critical, particularly for applications that prioritize scalability, efficiency, and where KGE methods still play a dominant role.
> > > >
> > > > [1] Glorot, X., & Bengio, Y. (2010, March). Understanding the difficulty of training deep feedforward neural networks. In Proceedings of the thirteenth international conference on artificial intelligence and statistics (pp. 249-256). JMLR Workshop and Conference Proceedings.
> > > >
> > > > [2] Chang, H., Ye, J., Lopez-Avila, A., Du, J., & Li, J. (2024, August). Path-based Explanation for Knowledge Graph Completion. In Proceedings of the 30th ACM SIGKDD Conference on Knowledge Discovery and Data Mining (pp. 231-242).
> > > >
> > > > [3] Ge, X., Wang, Y. C., Wang, B., & Kuo, C. C. J. (2024). Knowledge Graph Embedding: An Overview. APSIPA Transactions on Signal and Information Processing, 13(1).
> > > >
> > > > [4] Bang, D., Lim, S., Lee, S., & Kim, S. (2023). Biomedical knowledge graph learning for drug repurposing by extending guilt-by-association to multiple layers. Nature Communications, 14(1), 3570.
> > > >
> > > > [5] Li, N., Yang, Z., Wang, J., & Lin, H. (2024). Drug–target interaction prediction using knowledge graph embedding. iScience.
> > > >
> > > > Thank you once again for your valuable feedback, which has helped us improve our work.

---

> > > > > ### Comment · Reviewer_c8rP · 2024-11-21
> > > > >
> > > > > Thanks to the author for the reply, which addresses my questions. However, considering the limited scope of the paper, I would keep my score as a weak accept.

---

> > > > > > ### Author Response · Authors · 2024-11-30
> > > > > >
> > > > > > Dear Reviewer c8rP,
> > > > > >
> > > > > > We sincerely appreciate the time and effort you have dedicated to reviewing our work. Your thoughtful questions and feedback have been invaluable in helping us improve the quality of our research. On behalf of all the authors, we would like to extend our heartfelt gratitude for your contributions during this process.
> > > > > >
> > > > > > Warm regards,
> > > > > >
> > > > > > The Authors

---

### Official Review · Reviewer_wvH8 · 2024-11-03

**Soundness:** 2
**Presentation:** 3
**Contribution:** 2
**Rating:** 6
**Confidence:** 4

**Summary:**

This paper presents KGExplainer, a framework for generating explanations for knowledge graph completion (KGC) predictions by identifying synergistic paths. The work introduces a method to identify multiple synergistic paths for explaining KGC model predictions, moving beyond single-path approaches used in prior work. The authors develop a distillation-based evaluator for assessing explanation quality, providing a quantitative way to measure the effectiveness of the generated explanations. The technical core of the approach is a greedy search algorithm for finding explanatory paths within local subgraph structures. While the framework shows promise in generating more comprehensive explanations than existing methods, there are several aspects that warrant further investigation and improvement.

**Strengths:**

1. The paper demonstrates the importance of explanations in knowledge graph completion through concrete examples, such as competitor analysis (Alibaba-JD.com case) and drug interaction prediction (Acetaminophen-Warfarin case) shown in Figure 1. These examples clearly illustrate why single-path explanations are insufficient for complex predictions.
2. The paper introduces a novel distillation-based evaluator that achieves comparable performance to the original KGE models, as evidenced by the results in Table 1. For instance, KGExp-RotatE achieves 17.33% Hits@1 on OGB-biokg compared to RotatE's 17.12%, showing the evaluator's reliability.
3. The proposed method shows consistent improvements across multiple datasets, and the method is theoretically proven scalable (in Section 4.4, computation complexity is shown depend mainly on local parameters such as path length and number of paths rather than total graph size.)

**Weaknesses:**

1. While the paper introduces the concept of "synergistic paths" as a key contribution, it lacks rigorous theoretical analysis of what constitutes meaningful synergy between paths. The paper never formally defines what makes paths truly synergistic versus simply having multiple independent supporting paths. Besides, the greedy search strategy for finding synergistic paths lacks theoretical guarantees about its optimality. I recommend that the authors should formalize the concept of path synergy, perhaps through information theoretic measures that capture the mutual information between paths, or through formal analysis of how different path combinations contribute to prediction confidence. This would strengthen the theoretical foundation of the work beyond just empirical results.
2. The experimental evaluation doesn't sufficiently demonstrate real-world usefulness of the explanations. The human evaluation, while showing 83.3% accuracy, involved only academic participants (students/professors) rather than domain experts or real end users who would actually use such systems. The experiments lack evaluation of how these explanations help in practical tasks like error analysis or model debugging. There are no case studies demonstrating how synergistic paths provide better insights than single paths in real applications. To properly validate the method's practical value, the authors may conduct user studies with domain experts (like biomedical researchers for OGB-biokg), include case studies showing how the explanations help debug model errors, analyze explanation patterns for both correct and incorrect predictions, and demonstrate benefits through concrete application scenarios.

**Questions:**

1. You state that 'a fact is usually determined by the synergy of multiple reasoning chains', could you provide empirical evidence or analysis showing what percentage of predictions actually require multiple paths versus being explainable by a single path?
2. In Section 4.4 you provide theoretical complexity analysis showing the method scales with path length L and number N rather than graph size. However, the empirical results in Table 4 show significant runtime differences between datasets. Could you explain this apparent discrepancy and provide more details about what factors dominate the practical runtime?
3. Regarding the human evaluation showing 83.3% accuracy, could you provide more details about a) The criteria given to participants for judging explanation quality; b) Whether participants were shown the explanations simultaneously or sequentially; and c) If the 20 test cases were randomly selected or curated?

---

> ### Author Response · Authors · 2024-11-18
> **A Kind Reply to Dear Reviewer wvH8**
>
> We appreciate your detailed review and constructive suggestions. We address the concerns and questions below.
>
> **Q1.** Formalizing the concept of path synergy and providing empirical evidence or analysis showing what percentage of predictions actually require multiple paths.
>
> **Response:** **(1)** Formalizing the concept of synergistic paths from an information-theoretic perspective.
>
> Let:
> - <$h, r, t$> denotes a triple, respectively, forming the target prediction.
> - $\mathcal{P}=\{p_1,p_2,\ldots,p_n \}$ be a set of paths between $h$ and $t$.
>
> Two variables are considered synergic if their combined effect is greater than the sum of their individual effects [1]. We aim to quantify the *synergy* among paths in $\mathcal{P}$.
> 1. Quantifies the information a single path $p_i$ contributes to the prediction $t$:$$I(t; p_i) = H(t) - H(t | p_i),$$
> where $H(t)$ is the entropy of $t$, and $H(t|p_i)$ is the conditional entropy of $t$ given $p_i$.
> 2. Measures the information provided by a subset of paths $\mathcal{P}\_{synergy} \subset \mathcal{P}$:
>    $$I(t; \mathcal{P}\_{synergy}) = H(t) - H(t | \mathcal{P}\_{synergy}).$$
> 3. The paths in $\mathcal{P}\_{synergy}$ are synergistic if:
>    $$I(t;\mathcal{P}\_{synergy}) > \sum_{p_i \in \mathcal{P}\_{synergy}}I(t; p_i),$$
>    meaning the collective information from $\mathcal{P}\_{synergy}$ exceeds the sum of individual contributions.
> 4. The optimal subset $\mathcal{P}\_{synergy}^\*$ maximizes the *synergistic mutual information gain*:
>    $$\mathcal{P}\_{synergy}^\* = \arg\max_{\mathcal{P}' \subseteq \mathcal{P}} \left( I(t; \mathcal{P}')-\sum_{p_i \in \mathcal{P}'} I(t; p_i) \right).$$
>
> **(2)** Quantifying mutual information among synergistic paths in KGs is often complex and highly context-dependent, making it challenging to determine the proportion of predictions requiring multiple paths. In human evaluations of Family-rr, KGExplainer demonstrated superior performance in approximately 80% of cases compared to single-path exploration methods, suggesting experimental evidence of the importance of multiple paths. Overall, this is an intriguing topic for further investigation, which we plan to explore in greater depth.
>
> [1] Barrera, N. P. (2005). Principles: mechanisms and modeling of synergism in cellular responses. Trends in Pharmacological Sciences.
>
> **Q2.** Explain the significant runtime differences between datasets and provide more details about what factors dominate the practical runtime.
>
> **Response:** The substantial differences in runtime across datasets can largely be attributed to variations in these local parameters. Table 1 presents an analysis of the average node degree, path length, and path number within the enclosing subgraphs. Our results show a strong positive correlation between the number of paths and computation time, which aligns closely with our theoretical analysis.
> |Datasets|avg. degree|avg. path length|avg. path number|avg. time|
> |-|-|-|-|-|
> |Family-rr|3.902|2.828|4.1|1.32|
> |FB15k-237|9.357|4.195 |50.054 |16.81|
> |OGB-biokg|101.579|3.982 |457.234|157.28|
> |
>
> Table 1. The distribution of degree, path length, and path number over various KGs.
>
> Specifically, the computational complexity of KGExplainer primarily arises from two factors: (1) the greedy search within $g$, and (2) the re-training cost associated with $g$. We have discussed the complexity of the greedy search is largely determined by local parameters. The complexity of the re-training stage for different base KGEs, which is dominated by the size of subgraph $g$ (i.e., $O(\frac{LN|\mathcal{V}|}{|\mathcal{E}|})$ discussed in Section 4.4), the embedding dimension $d$, and the size of negative samples $n$. We can analyze the efficiency of base KGEs as follows:
> + In TransE, DistMult, and RotatE, the computational efficiency of each fact within $g$ is $O((n+1)\*d)$. Therefore, the overall complexity is $O(\frac{LN|\mathcal{V}|}{|\mathcal{E}|}\*(n+1)\*d)$, where $L$ is the maximum path length, $N$ denotes the path number.
>
> In conclusion, the practical runtime largely depends on the local parameters of the knowledge graphs.
>
> **Q3.** Provide more details about the human evaluation.
>
> **Response:** Let us address your concerns point by point:
>
> a) We give two bases for judgment as a guide when asking participants questions.
> 1. In Family-rr, for predicting the kinship between two individuals, the explanation should be based on the known family memberships, marital relationships, etc. reasonably derived.
> 2. Encourage testers to choose more concise and clear explanations, while maintaining completeness. Avoid overly complex explanations so that key information can be presented clearly.
>
> b) The results from different models are presented to the participants simultaneously.
>
> c) We randomly selected 20 samples from the test set of Family-rr for participants to evaluate.
>
> Once again, we appreciate your detailed feedback and suggestions, which have strengthened the presentation and robustness of our work.

---

### Official Review · Reviewer_PA5a · 2024-11-04

**Soundness:** 3
**Presentation:** 3
**Contribution:** 3
**Rating:** 6
**Confidence:** 4

**Summary:**

The paper presents KGExplainer, a model-agnostic framework designed to provide path-based explanations for knowledge graph completion. The approach leverages a perturbation-based greedy search algorithm to identify the most important synergistic paths that contribute to the prediction of facts within a knowledge graph. The method is grounded in the insight that predictions are often supported by multiple interconnected reasoning chains. To evaluate these explanations, KGExplainer uses a distilled evaluator derived from the target knowledge graph embedding model, ensuring the explanations maintain high fidelity.

**Strengths:**

- Model-Agnostic Approach. KGExplainer is designed to work with different KGE models, making it a versatile tool for enhancing explainability across various KGC tasks.
- Theoretical Justification. The paper provides a solid theoretical foundation for the proposed framework, in terms of the scalability of KGExplainer.

**Weaknesses:**

- Limited Scalability Experiments and Discussion. While the paper mentions that KGExplainer can handle large KGs with small average degrees, there is limited discussion on how the framework scales with increasing graph size and complexity.

**Questions:**

Please see weaknesses.

---

> ### Author Response · Authors · 2024-11-18
> **A Kind Reply to Dear Reviewer PA5a**
>
> Thank you for your detailed review and constructive suggestions. We provide a detailed discussion of the reviewer's concerns.
>
>
> **1.** Limited discussion on how the framework scales with increasing graph size and complexity.
>
> **Response:**
> To improve scalability for larger KGs, we can adjust the local enclosing subgraph size (e.g., reducing from a 2-hop to a 1-hop subgraph) to decrease path length and path count. Table 1 presents the distribution of node degree, path length, path number, and computation time across different subgraph scales over OGB-biokg, revealing a strong positive correlation between them, consistent with our theoretical analysis.
> |Scale|avg. degree|avg. path_length|avg. path number|avg. time|
> |-|-|-|-|-|
> |1-hop|97.265|2.573|346.331|103.245|
> |2-hop|101.579|3.982|457.234|157.28|
> |
>
> Table 1. The distribution of average degree, path length, path number, and computation time over various subgraph scales.
>
> Specifically, the computational complexity of KGExplainer primarily arises from two factors: (1) the greedy search within $g$, and (2) the re-training cost associated with $g$. As discussed in Section 4.4, the complexity of the greedy search is largely determined by local parameters (such as the maximum length of paths $L$). The re-training cost, on the other hand, depends on the underlying KGE model, which is dominated by the size of subgraph $g$ (i.e., $O(\frac{LN|\mathcal{V}|}{|\mathcal{E}|})$ discussed in Section 4.4), the embedding dimension $d$, and the size of negative samples $n$. Thus we can analyze the efficiency of base KGEs as follows:
> + In TransE, DistMult, and RotatE, the computational efficiency of each fact within $g$ is $O((n+1)\*d)$. Therefore, the overall complexity is $O(\frac{LN|\mathcal{V}|}{|\mathcal{E}|}\*(n+1)\*d)$, where $L$ is the maximum path length, $N$ denotes the path number.
>
> In conclusion, the complexity of KGExplainer is driven primarily by local parameters—such as path length $L$, path number $N$, and average degree—rather than the overall size of the knowledge graph (KG).
>
> Thank you again for your valuable insights, which will help us further strengthen the presentation and impact of our work.

---

> > ### Comment · Reviewer_PA5a · 2024-11-21
> >
> > Thanks for the authors' response. The experimental results and discussion makes sense to me on justifying the capability of KGExplainer for large KGs with small average degrees.
> > I would like to keep my scores since they are already positive.

---

> > > ### Author Response · Authors · 2024-11-30
> > >
> > > Dear Reviewer PA5a,
> > >
> > > We're happy to address your concerns, which further enhance the presentation of our work. On behalf of all the authors, I would like to extend our sincerest greetings and express our heartfelt gratitude for your dedicated time and effort during the review process.
> > >
> > > Warm regards,
> > >
> > > The Authors

---

### Comment · Area_Chair_iNem · 2024-11-21
**Please initiate discussion based on the rebuttal.**

Dear Reviewers,

The authors have posted a rebuttal to the concerns raised. I would request you to kindly go through their responses and discuss how/if this changes your opinion of the work. I thank those who have already initiated the discussion.

best,

Area Chair

---

### Meta-Review · Area_Chair_iNem · 2024-12-06

**Metareview:**

The paper proposes an algorithm for explaining knowledge graph embedding methods. It employs a perturbation-based greedy search algorithm to identify the key synergistic paths that are instrumental in predicting facts within a knowledge graph. After the rebuttal phase, all reviewers have expressed support for the paper, despite some reservations regarding its limited scope, especially considering that GNN-based methods have outperformed KGE methods.

**Additional Comments On Reviewer Discussion:**

The paper received three reviews. Two reviewers raised issues regarding its empirical comprehensiveness, while another highlighted a lack of theoretical grounding. The authors responded to these concerns in their rebuttal, after which all reviewers agreed to support the paper with a score of 6. Reviewer c8rP questioned the novelty, noting that knowledge graph embedding methods have been largely overtaken by GNN-based methods. I concur with this observation.

---

### Decision · Program_Chairs · 2025-01-22

Accept (Poster)